# ONLINE FINETUNING DECISION TRANSFORMERS WITH POLICY GRADIENTS

## ABSTRACT

Decision Transformer (DT) has emerged as a powerful paradigm for decision making by formulating offline Reinforcement Learning (RL) as a sequence modeling problem. While recent studies have started to investigate how Decision Transformers can be extended to online settings, online finetuning with pure RL gradients remains largely underexplored: most existing approaches continue to prioritize supervised sequence modeling losses during the online phase. We identify hindsight return relabeling—a component widely used in online DTs—as a key obstacle that, while beneficial for supervised objectives, hinders the performance of importance sampling-based RL algorithms such as PPO and GRPO. In this work, we present a new algorithm that enables online finetuning of Decision Transformers purely with reinforcement learning gradients. Our approach represents a novel adaptation of the classical GRPO algorithm to the online finetuning of Decision Transformers. To make GRPO efficient and compatible with DTs, we incorporate several key modifications, including sub-trajectory sampling, sequence-likelihood objectives, and an active sampling strategy. We conduct extensive experiments across diverse benchmarks and show that, on average, our method significantly outperforms existing online finetuning approaches such as ODT and ODT+TD3. This opens a new direction for advancing the online finetuning of Decision Transformers.

## 1 INTRODUCTION

Transformers (Vaswani et al., 2017) have become the dominant architecture across a wide range of domains. In large language models (LLMs), a powerful training paradigm has emerged: supervised pretraining on large-scale unlabeled corpora, followed by finetuning and reinforcement learning (Radford et al., 2018; Brown et al., 2020; Ouyang et al., 2022). Inspired by this success, Decision Transformer (DT) (Chen et al., 2021) introduced the transformer architecture into decision making problems, offering a new approach that formulates RL as sequence modeling. Unlike conventional RL methods, DT is trained entirely offline with a supervised objective on collected trajectories, effectively functioning as a variant of imitation learning (Hussein et al., 2017) conditioned on a pre-specified value of the initial return-to-go (RTG).

Its online variant, ODT (Zheng et al., 2022), further extended this approach by enabling online finetuning after pretraining. ODT collects online trajectories and use *hindsight return relabeling*, replacing the (pre-specified) RTGs of the online trajectories with the actual achieved returns. The purpose of this hindsight return relabeling is to align the RTG distribution of online trajectories with that of the offline dataset, since both offline pretraining and online finetuning optimize the same sequence modeling objective. And recent work augmenting it with TD3 (Fujimoto et al., 2018) gradients to achieve state-of-the-art performance (Yan et al., 2024). However, existing approaches to online finetuning of DT remain dominated by supervised objectives: ODT relies solely on supervised loss, while ODT+TD3 combines it with TD3 gradients. Yet, recent breakthroughs in LLMs demonstrate that purely reinforcement learning gradients like Proximal Policy Optimization (PPO) and Group Relative Policy Optimization (GRPO) can fundamentally enhance a transformer's reasoning capabilities (Shao et al., 2024; Team, 2025; Yang et al., 2024). This trend naturally raises a natural and critical question:

*Can we conduct online finetuning of Decision Transformers with pure RL gradients?*

To investigate this question, we first revisit the training paradigm of existing online variants of DT and uncover a core challenge. We find that hindsight return relabeling deployed by existing online variants such as ODT and ODT+TD3 actually hinders the application of on-policy RL algorithms that rely on importance sampling. Specifically, hindsight return relabeling introduces a critical mismatch between the return-to-go during online interaction and the training phase, which ultimately impairs model performance. Removing this component is the necessary premise for applying importance sampling based algorithms to DTs as shown in Fig. 1a.

Building on this key insight, we develop a new algorithm for online finetuning DTs with pure RL gradients. Specifically, we adapt GRPO, an algorithm that has demonstrated remarkable effectiveness in LLM reasoning, to the characteristics of conventional RL environments. Our method incorporates several critical modifications: (1) a sub-trajectory based training objective that mitigates estimation variance and improves credit assignments—an aspect known to be challenging for standard GRPO; (2) environment resetting techniques (Mhammedi et al., 2024; Yin et al., 2023) to provide consistent initial states; (3) sequence-level importance ratios that enhance efficiency and stability; (4) active selection that encouraging exploration where the policy is uncertain. With the above adaptations, our GRPO achieves state-of-the-art performance in online finetuning of Decision Transformers. Moreover, for scenarios where environment resetting is not feasible, training an auxiliary Q-function to substitute the resetting process still yields decent results. Additionally, we also apply PPO to DTs, showing its ability to improve pretrained DTs as well.

Our adapted GRPO achieves higher rewards, requires no auxiliary critic, and is more computationally efficient as it requires much less gradient updates compared to previous methods. Moreover, unlike methods such as ODT+TD3 that modify the pretraining loss and train an extra Q-function while pretraining, our approach can directly finetune most pretrained DT-style models with minimal changes (see Appendix A.5 for experiments).

**Contributions.**   We summarize our main contributions below:

(i) We identify hindsight return relabeling as the key obstacle that prevents effective finetuning of Decision Transformers with PPO/GRPO.

(ii) We introduce GRPO-DT, an adaptation of GRPO for Decision Transformers that integrates sub-trajectory optimization, sequence-level importance ratios, and active state sampling, enabling pure-RL finetuning of Decision Transformers.

(iii) We conduct extensive experiments and show that online finetuning DT with pure RL gradients can achieve new state-of-the-art results on several benchmarks.

**Paper organization.**   The rest of the paper is organized as follows. Section 2 reviews preliminaries on DT, GRPO and related concepts. Section 3 elaborates our proposed method. Section 4 presents experiments and results. Section 5 and Section 6 provide related work and conclude paper respectively.

## 2  PRELIMINARIES

**Markov Decision Process.**   We formulate the reinforcement learning problem as a *Markov Decision Process* (MDP), defined by a tuple $(\mathcal{S}, \mathcal{A}, P, R, \gamma)$. Here, $\mathcal{S}$ is the state space, $\mathcal{A}$ is the action space, $P(s_{h+1} \mid s_h, a_h)$ denotes the transition dynamics, $R(s_h, a_h)$ is the immediate reward, and $\gamma \in [0, 1]$ is the discount factor. At each step $h = 1, \ldots, H$, the agent observes $s_h \in \mathcal{S}$ and selects an action $a_h \in \mathcal{A}$ according to a policy, either stochastic $\pi(a_h|s_h)$ or deterministic $\mu(s_h)$. The environment then transitions to $s_{h+1} \sim P(\cdot|s_h, a_h)$ and yields a reward $r_h = R(s_h, a_h)$. A trajectory is thus $(s_1, a_1, r_1, \ldots, s_H, a_H, r_H)$, and the objective of reinforcement learning is to maximize the expected discounted return $\mathbb{E}_\pi \left[ \sum_{h=1}^H \gamma^{h-1} r_h \right]$.

**Decision Transformers.**   Decision Transformer (DT) is a powerful paradigm for offline reinforcement learning, formulating decision making as a sequence modeling problem. Instead of relying on temporal-difference errors, DT reframes offline RL into a supervised learning setting. A DT sequence consists of three types of tokens: *return-to-go* (RTG), *state*, and *action*. The RTG at step $h$, denoted $g_h$, represents the cumulative reward from step $h$ onward. DT leverages a GPT-style architecture

(Radford et al., 2018) to autoregressively learn a deterministic policy from pre-collected trajectories. In practice, DTs are trained on fixed-length trajectory segments rather than full episodes: let $K$ denotes the context length, the DT learns to generate the next action $a_h$ based on past interactions $(g_{h-K+1}, s_{h-K+1}, a_{h-K+1}, \ldots, g_h, s_h)$ of context length $K$. The model is trained via supervised learning by minimizing the mean squared error (MSE) between the predicted action and the ground-truth action. During evaluation and deployment, the learner specifies a desired initial RTG $g_0$, since the ground-truth future RTG isn't known in advance, and leverages the DT to autoregressively generate the next action and interact with the environment.

**Online Finetuning of Decision Transformers.** ODT extends DT into the **online setting**. After pretraining, it continues training while interacting with the environment, collecting trajectories that gradually replace the offline buffer. ODT learns a stochastic Gaussian policy conditioned on past returns, states, and actions:

$$\pi_\theta(a_h \mid \mathbf{s}_{-K,h}, \mathbf{g}_{-K,h}, \mathbf{a}_{-K,h-1}) = \mathcal{N}\big(\mu_\theta(\mathbf{s}_{-K,h}, \mathbf{g}_{-K:h}, \mathbf{a}_{-K,h-1}), \Sigma_\theta(\mathbf{s}_{-K,h}, \mathbf{g}_{-K,h}, \mathbf{a}_{-K,h-1})\big)$$

where $\theta$ denotes the policy parameters, $\Sigma_\theta$ is the diagonal covariance matrix, $-K, h$ means past $K$ steps before $h$. However, Yan et al. (2024) pointed out that because ODT models actions conditioned on desired returns, it actually learns $\frac{\partial a}{\partial \text{RTG}}$: how actions change as the target return varies. However, what drives online policy improvement is $\frac{\partial \text{RTG}}{\partial a}$: how returns respond to action adjustments (see section 3.1 in Yan et al. (2024) for more details). Yan et al. (2024) thus propose ODT+TD3, which augments ODT loss with TD3 gradients to provide $\frac{\partial \text{RTG}}{\partial a}$ to guide online exploration, which is particularly crucial when the offline dataset is of low quality. However, they still prioritize supervised ODT loss as their main training objective.

**Group Relative Policy Optimization (GRPO).** GRPO is initially proposed in DeepSeek-math(Shao et al., 2024) for Large Language Models(LLMs) post-training. It bypasses the need for value model by computing the relative advantage of each response within a group of responses given the same query. Specifically, the model generates a group of responses $o_0, o_1, \ldots, o_G$ from the old policy $\pi_{\theta_{\text{old}}}$ for each question $q$ sampled from the question set $Q$. For each response $o_i$, a reward $r_i$ is specified. Then the policy model is optimized by maximizing the following objective:

$$J_{\mathsf{GRPO}}(\pi_\theta) = \mathbb{E}_{q \sim Q, \{\alpha_i\}_{i=1}^G \sim \pi_{\theta_{\text{old}}, i \in I}^{(l)}}$$

$$\left\{ \frac{1}{G} \sum_{i=1}^G \frac{1}{|\alpha_i|} \sum_{h=1}^{|\alpha_i|} \min \left\{ w_{i,h}(\theta) \hat{A}_i, \mathsf{clip}\left(w_{i,h}(\theta), 1-\varepsilon, 1+\varepsilon\right) \hat{A}_i - \beta D_{\mathsf{KL}}\left[\pi_\theta \parallel \pi_{\mathsf{ref}}\right] \right\} \right\}, \quad (1)$$

where $G$ is the number of generated responses to each query $q$, importance ratio $w_{i,h}(\theta) = \frac{\pi_\theta(\alpha_{i,h} \mid q, \alpha_{i,<h})}{\pi_{\theta_{\text{old}}}(\alpha_{i,h} \mid q, \alpha_{i,<h})}$ and the advantage of $i$-th rollout $\hat{A}_i = \frac{r_i - \text{mean}(\{r_1, r_2, \cdots, r_G\})}{\text{std}(\{r_1, r_2, \cdots, r_G\})}$.

## 3 METHODS

This section is organized as follows. We first analyze the limitations of prior attempts at online finetuning Decision Transformers with importance sampling based algorithms (e.g., PPO/GRPO) and present our solutions. Based on this we describe our adaptation of GRPO to reinforcement learning environments, highlighting several key modifications to naive GRPO.

### 3.1 REMOVING HINDSIGHT RETURN RELABELING

When deploying DTs, the learner must specify a desired initial RTG, since the ground-truth future RTG is unknown in advance. In Online Decision Transformer (ODT), the learner typically sets a relatively high target RTG $g_{\mathsf{high}}$ during rollout to encourage optimistic exploration. During training, a key component of ODT—known as hindsight return relabeling—replaces the intended RTG $g_{\mathsf{high}}$ with the actually achieved RTG $g_{\mathsf{relabel}}$ (Zheng et al., 2022).

This hindsight return relabeling, while necessary for sequence modeling to align the distribution of RTG from offline dataset with online trajectories (see Fig. 5.4 in Zheng et al. (2022) for details),

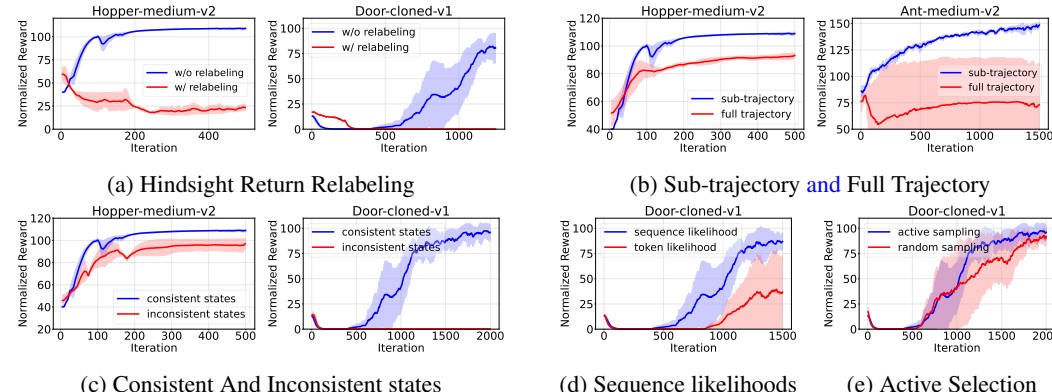

(a) Hindsight Return Relabeling  (b) Sub-trajectory and Full Trajectory

(c) Consistent And Inconsistent states  (d) Sequence likelihoods  (e) Active Selection

Figure 1: Examples of GRPO with and without some of our key designs. (a) compares reward curves with and without hindsight return relabeling when processing sampled sub-trajectories. (b) compares the learning process of our adapted GRPO (using sub-trajectories) against naive GRPO (using complete trajectories). (c) shows the effect of using consistent states when sampling a group versus not. (d) illustrates the difference in the learning process when computing the importance ratio using sequence likelihood versus token likelihood. (e) compares the learning process with and without active selection for sampling reset points.

actually introduces inconsistencies of RTGs between rollout and training phases, which hinder effective on-policy optimization. This means the policy generates actions conditioned on an optimistic RTG target during rollout, but the same actions are later trained against a trajectory labeled with the actually achieved (and often smaller) RTG. This creates a discrepancy in the conditioning variable: the policy is effectively asked to maximize likelihood under goals it never explicitly conditioned on during execution. Actions are drawn under $\pi_{\text{old}}(a|s, g_{\text{high}})$ but later trained as if they came from $\pi_{\text{old}}(a|s, g_{\text{relabel}})$, the importance weights then become unreliable, undermining stable on-policy optimization. This also explains why naive attempts at applying standard PPO to ODT fails in Yan et al. (2024) (Appendix C in their paper).

To address this, one must carefully align rollout conditioning with training objectives. In our practice, we simply store the intended RTGs alongside each trajectory to preserve consistency. Our ablation experiments in Fig. 1a demonstrate that without such modification, applying importance sampling based algorithms to ODT remains unstable. In relatively simple environments such as Hopper, the policy may initially improve but eventually collapses. In more complex environments such as Door, the policy fails to learn altogether.

### 3.2 ADAPTING GRPO TO DECISION TRANSFORMERS

Our method adapts GRPO to Decision Transformers by training on sub-trajectories instead of full trajectories used in original GRPO. At each iteration, the policy interacts with the environment to collect full trajectories, from which we sample reset points and generate groups of sub-trajectories under corresponding conditions. The sub-trajectories within the same group are then assigned normalized advantages with Eq. (2). These sub-trajectories and their advantages are finally used to update the policy with Eq. (3). This also aligns the finetuning process with the sub-trajectory modeling paradigm when pretraining DTs. The details of our training pipeline are described in Algorithm 1

Compared to the vanilla GRPO, our method introduces four key design modifications to better align with the Decision Transformer framework and continuous control setting. Specifically, (i) we redesign the optimizing objective by operating on sub-trajectories rather than full rollouts, (ii) we ensure the consistency of initial states when generating sub-trajectories by resetting environments to the same corresponding state (iii), we compute importance weights at the sequence level to match the unit of reward, and (iv) we incorporate an active selection mechanism that prioritizes uncertain states for optimization. We elaborate on each of these design choices below.

---

**Algorithm 1** Decision Transformers with GRPO (GRPO-DT)

---

**Input:** Pretrained policy $\pi_\theta$, complete trajectory buffer $\mathcal{T}_{\mathsf{replay}}$, sub-trajectory buffer $\mathcal{T}_{\mathsf{sub}}$, expected initial RTG $g_0$, total rounds $T$, number of reset points in a trajectory $K$, sub-trajectory length $L_{\mathsf{traj}}$, evaluation steps $L_{\mathsf{eval}}$, group size $G$, discount factor $\gamma$.

1: **for** round $t = 1, \cdots, T$ **do**
2:  Rollout complete trajectory $\tau$ using current policy $\pi_\theta(\cdot|s_0, g_0)$, conditioned on initial state $s_0$ and RTG $g_0$; update $\mathcal{T}_{\mathsf{replay}}$ with $\tau$). `// Collect complete policy; buffer updated in a FIFO manner.`
3:  Sample a minibatch $\mathcal{G}$ from $\mathcal{T}_{\mathsf{replay}}$ with probability $p(\tau) = \frac{|\tau|}{\sum_{\tau \in \mathcal{T}} |\tau|}$.
4:  **for** each $\tau \in \mathcal{G}$ **do**
5:   Sample $K$ reset points $\{s_{h_k}\}_{k=1}^K$ from action-variance distribution.
6:   For each reset point $s_{h_k}$, generate $G$ sub-trajectories $\{\tau_{k_i}^{\mathsf{sub}}\}_{i=1}^G$ with the current policy $\pi_{\theta_t}$; evaluate the quality of each sub-trajectory to get reward $R(\tau_{k_i}^{\mathsf{sub}})$. `// Sub-trajectory generation and evaluation.`
7:   Compute advantage $\widehat{A}(\tau_{k_i}^{\mathsf{sub}})$ for each sub-trajectory using Eq. (2). `// Compute advantages for GRPO.`
8:   Update sub-trajectory buffer $\mathcal{T}_{\mathsf{sub}}$ with $\{(\tau_{k_i}^{\mathsf{sub}}, \widehat{A}(\tau_{k_i}^{\mathsf{sub}}))\}_k$ `// Buffer updated in a FIFO manner.`
9:  Finetune the current policy with sub-trajectories in $\mathcal{T}_{\mathsf{sub}}$ and Eq. (3) to get a new policy $\pi_{\theta_{t+1}}$.

---

**(1) Optimization on sub-trajectories.** In its original formulation to train LLMs, GRPO assigns a single response-level reward to each generated response, with every token sharing the same reward. A direct adaptation to continuous control problems would be to aggregate all stepwise rewards in a rollout and assign advantages computed based on this trajectory-level return to each step, but this method leads to poor performance (Fig. 1b). This limitation is expected, as reinforcement learning tasks—particularly those in continuous control—require more precise credit assignment than language modeling. Whereas tokens in a sentence tend to be coherently correlated, actions in RL can lead to drastically different outcomes (e.g., distinct action choices when navigating a maze).

To address this, we adopt a sub-trajectory formulation: from the policy's action distribution we sample a segment of length $L_{\mathsf{traj}}$, and then continue the rollout deterministically by taking the mean action (or the most probable action in the discrete case) for another $L_{\mathsf{eval}}$ steps. The cumulative discounted reward over these $L_{\mathsf{traj}} + L_{\mathsf{eval}}$ steps is attributed to the preceding sub-trajectory and then used to compute advantages within a group with Eq. (2).

$$\widehat{A}_{k_i} = \frac{r_{k_i}^{\mathsf{sub}} - \mathrm{mean}(\{r_{k_1}^{\mathsf{sub}}, r_{k_2}^{\mathsf{sub}} \cdots, r_{k_{|G|}}^{\mathsf{sub}}\})}{\mathrm{std}(\{r_{k_1}^{\mathsf{sub}}, r_{k_2}^{\mathsf{sub}} \cdots, r_{k_{|G|}}^{\mathsf{sub}}\})}. \tag{2}$$

Only the sub-trajectory of length $L_{\mathsf{traj}}$ is used for GRPO optimization, while the subsequent $L_{\mathsf{eval}}$ steps are used solely for evaluation. The parameter $L_{\mathsf{traj}}$ controls the granularity of credit assignment, whereas $L_{\mathsf{eval}}$ determines the quality of reward estimation. Empirically, we find that a smaller $L_{\mathsf{traj}}$ combined with a larger $L_{\mathsf{eval}}$ yields the best performance; see Section 4.3 for detailed ablations on these hyperparameters.

**(2) Providing consistent states.** GRPO requires rollouts within the same group to be conditioned on the same prompt, which in continuous control corresponds to starting from the same environment state. If sub-trajectories originate from different states but are grouped together when computing advantages with Eq. (2), their returns become incomparable and training fails to converge as shown in Fig. 1c. We therefore enforce state consistency by resetting vectorized environments to specified states before generating sub-trajectories. This reset mechanism is crucial for stable optimization.

Environment resets are supported in many important domains—including perfect-information games (e.g., Go, Chess), LLM reasoning tasks (Kazemnejad et al., 2024), and widely used simulator-based RL benchmarks (Mhammedi et al., 2024). Recent theoretical and empirical work also shows that incorporating reset operations can substantially improve sample efficiency and policy performance in online RL (Mhammedi et al., 2024; Yin et al., 2023). Our method follows this established line of work. In scenarios where resetting is infeasible, we find that evaluating multiple candidate actions

under the same state with a learned Q-function that is trained following TD3 (Fujimoto et al., 2018), and applying GRPO at action level (see Appendix A.4 for details) yields decent results.

**(3) Sequence-level importance ratio.** In naive GRPO, importance weights are computed at the token level, reflecting stepwise likelihoods. However, in our setting advantages are defined for the entire sub-trajectories, making token-level ratios misaligned with the unit of reward. This motivates us to forego the token-level objective and explore utilizing importance weights and performing optimization at the sequence level. We therefore compute importance ratios directly on sub-trajectories with Eq. (3), ensuring consistency between the objective and the advantage signal. Note that Eq. (1) and Eq. (3) differ primarily in their optimization granularity: the former operates at the token level, whereas the latter is defined at the sequence level. This sequence-level importance ratio improves both stability and efficiency as shown in Fig. 1d. This is in line with the concurrent work (Zheng et al., 2025).

$$
J_{\text{GRPO}}(\theta) = \frac{1}{N} \sum_{i=1}^{N} \left\{ \min \left[ \frac{\pi_{\theta_t}(\tau_i^{\text{sub}}|s_{i,0}, g_{i,0})}{\pi_{\theta_{old}}(\tau_i^{\text{sub}}|s_{i,0}, g_{i,0})} \widehat{A}_i, \text{clip} \left( \frac{\pi_{\theta_t}(\tau_i^{\text{sub}}|s_{i,0}, g_{i,0})}{\pi_{\theta_{old}}(\tau_i^{\text{sub}}|s_{i,0}, g_{i,0})}, 1-\varepsilon, 1+\varepsilon \right) \widehat{A}_i \right] \right.
$$
$$
\left. - \beta \, \mathbb{D}_{KL}[\pi_{\theta_t}||\pi_{\text{ref}}] \right\} + \kappa \, \mathcal{H}_\theta(\mathbf{a}|\mathbf{s}, \mathbf{g}). \tag{3}
$$

where $\mathbb{D}_{KL}[\pi_{\theta_t}||\pi_{\text{ref}}] = \frac{\pi_{\text{ref}}(\tau_i^{\text{sub}}|s_{i,0}, g_{i,0})}{\pi_{\theta_t}(\tau_i^{\text{sub}}|s_{i,0}, g_{i,0})} - \log \frac{\pi_{\text{ref}}(\tau_i^{\text{sub}}|s_{i,0}, g_{i,0})}{\pi_{\theta_t}(\tau_i^{\text{sub}}|s_{i,0}, g_{i,0})} - 1$ is the KL-penalty, and $\mathcal{H}_\theta(\mathbf{a}|\mathbf{s}, \mathbf{g})$ denotes the entropy regularization term. Following ODT, its coefficient $\kappa$ is treated as a trainable parameter to better balance exploration and exploitation.

**(4) Active selection.** During action generation, we observe that certain timesteps exhibit high variance in the predicted action distribution. When sampling actions, this variance leads to diverse generated actions, suggesting that the policy is uncertain about which action to take. As a result, improving behavior specifically on these states is beneficial and aligns with prior findings showing that prioritizing uncertain regions can accelerate policy improvements (Yin et al., 2023). at these steps. To address this, we introduce a simple yet effective technique called active selection. Concretely, for a given complete trajectory, we apply a softmax transformation to the action variance sequence across timesteps using $p_t = \frac{\exp(\sigma_t^2)}{\sum_{k=0}^{|\tau|} \exp(\sigma_k^2)}$ to yield a probability distribution. We then sample reset points from this distribution to determine where to initiate sub-trajectory generation. Empirically, as shown in Fig. 1e, our active selection mechanism outperforms variants without it.

## 4 EXPERIMENT

In this section, we aim to answer the following questions:

(i) Do pure RL gradients provide better signals compared with methods that prioritize supervised loss during DT online finetuning?

(ii) How does each component in our method affect the performance?

The model architecture and hyperparameter setting can be found in Appendix A.3.1.

### 4.1 EXPERIMENT SETUP

**Tasks and datasets.** We evaluate methods on three continuous control and manipulation environments from D4RL (Fu et al., 2020): (i) **MuJoCo** (Todorov et al., 2012) tasks, including *Hopper*, *Walker2d*, and *Ant*, with dense rewards, evaluated on the *medium*, *medium-replay*, and *random* datasets. (ii) **Adroit** manipulation tasks (Rajeswaran et al., 2017), including *Door*, *Hammer*, and *Pen*, evaluated on the *human* and *cloned* datasets. (iii) **Antmaze** (Fu et al., 2020) with sparse goal-reaching rewards (a reward of 1 if success and 0 otherwise), using the *umaze* and *umaze-diverse* datasets. Detailed descriptions of each environment and dataset are provided in Appendix A.1.

**Baselines.** In our experiments, we mainly compare both our adapted GRPO-DT and PPO-DT with three baselines: **ODT** (Chen et al., 2021), the widely adopted online version of Decision Transformer

Table 1: Average reward for each method. The best performance and results > 99% of the best result is bold. Results > 90% of the best result are underlined. The name of the environments and datasets are abbreviated as follows: Ho=Hopper, Wa=Walker2d, An=Ant, U=Antmaze-umaze, UD=Antmaze-umaze-diverse, D=Door, P=Pen, H=Hammer; for the datasets M=Medium, MR=Medium-Replay, R=Random, C=Cloned, H=Human. The format is "final (standard deviation)".

|  |  | DT | IQL | ODT | TD3+ODT | PPO-DT | GRPO-DT |
|---|---|---|---|---|---|---|---|
| MuJoCo (random) | Ho-R-v2 | 1.98 | 41.02 (13.35) | 30.43 (0.01) | 83.32 (8.46) | **106.97 (0.96)** | 99.20 (3.80) |
|  | Wa-R-v2 | 4.59 | 22.75 (1.55) | 10.88 (0.34) | 82.95 (18.28) | **108.69 (8.86)** | 100.25 (33.19) |
|  | An-R-v2 | 30.38 | 58.69 (23.03) | 19.08 (3.97) | 80.58 (7.25) | 107.45 (22.83) | **120.69 (5.47)** |
|  | Average | 12.32 | 40.82 | 20.13 | 82.28 | **107.70** | 106.71 |
| MuJoCo (medium) | Ho-M-v2 | 63.1 | 74.19 (20.25) | 98.02 (0.63) | 101.47 (2.29) | 105.65 (5.43) | **108.81 (0.85)**, |
|  | Ho-MR-v2 | 29.76 | 96.97 (2.16) | 87.73 (0.59) | **107.94 (2.29)** | **109.60 (1.63)** | 83.61 (20.75) |
|  | Wa-M-v2 | 70.78 | 103.45 (1.37) | 76.49 (0.78) | 103.27 (5.95) | 109.49 (9.04) | **158.34 (3.75)** |
|  | Wa-MR-v2 | 58.06 | 103.00 (2.65) | 74.21 (2.41) | 102.80 (2.68) | 117.45 (14.79) | **137.36 (5.64)** |
|  | An-M-v2 | 90.58 | 118.18 (2.42) | 90.71 (0.03) | 131.56 (0.41) | 139.84 (0.95) | **147.51 (2.44)** |
|  | An-MR-v2 | 78.15 | 117.51 (0.82) | 83.63 (0.87) | 120.01 (2.94) | 117.95 (2.54) | **142.05 (3.32)** |
|  | Average | 65.07 | 102.21 | 85.13 | 111.175 | 116.66 | **129.61** |
| Adroit | D-C-v1 | 4.97 | 46.72 (0.30) | 1.26 (1.02) | 79.98 (5.62) | 0.19 (0.00) | **96.41 (7.59)** |
|  | D-H-v1 | 9.30 | 11.27 (0.44) | 8.76 (3.87) | 79.73 (4.37) | **94.12 (3.99)** | 89.33 (10.12) |
|  | P-C-v1 | 75.02 | 63.09 (14.38) | 16.24 (5.12) | **109.86 (6.27)** | 27.14 (0.24) | **111.15 (2.61)** |
|  | P-H-v1 | 95.23 | 24.94 (1.48) | 19.84 (7.42) | 77.18 (7.42) | 9.92 (5.00) | **85.11 (6.08)** |
|  | H-C-v1 | 1.80 | 4.87 (3.10) | 1.32 (0.06) | 119.95 (2.45) | 130.60 (2.81) | **140.45 (1.93)** |
|  | H-H-v1 | 1.01 | 1.04 (1.56) | 0.91 (0.22) | 120.93 (2.18) | 129.23 (2.18) | **132.64 (12.56)** |
|  | Average | 31.22 | 25.15 | 8.06 | 97.93 | 65.2 | **109.18** |
| Antmaze | U-v2 | 16.00 | 91.21 (2.14) | 89.27 (3.73) | **99.64 (0.20)** | 0.00 (0.00) | 96.07 (0.53) |
|  | UD-v2 | 38.00 | 0.00 (0.00) | 63.81 (1.64) | **99.42 (0.43)** | 47.00 (4.00) | 97.70 (2.67) |
|  | Average | 27 | 45.61 | 76.54 | **99.53** | 23.50 | 96.89 |

with supervised loss as online finetuning objective; **ODT+TD3** (Yan et al., 2024), the current state-of-the-art method for online finetuning of Decision Transformer; **IQL** (Kostrikov et al., 2021), a popular offline algorithm which also has an online variant.

**Metrics.** We use the normalized average reward of 3 random seeds according to D4rl's statistic (Fu et al., 2020) where higher rewards represent better performance. Meanwhile, we also present the learning curves which shows the change of the normalized rewards with respect to the training iterations. When presenting the curves, we set the x-coordinate to be the number of iteration. This variable is the *round* from line 3 of the Algorithm. 1 from ODT Zheng et al. (2022) paper. Note that conventional x-axis metrics, such as the number of online transitions (indicating sample efficiency) and the number of gradient updates (indicating computational cost), are not suitable for our setting. For gradient updates, ODT/ODT+TD3 requires nearly two orders of magnitude more updates per iteration compared to our PPO-DT/GRPO-DT; for online interactions, our GRPO-DT and PPO-DT consume several to tens of times more samples than ODT/ODT+TD3. Hence, neither metric provides a fair comparison. When evaluate, we conduct evaluation after the gradient updates of the corresponding iteration. Thus, even at iteration 0, all methods have already undergone several updates, during which their behaviors may diverge and produce different outcome.

**PPO-DT implementation.** Our PPO-DT implementation follows the practice of CleanRL (Huang et al., 2022). Unlike prior work that applies PPO to multi-agent reinforcement learning (MARL) tasks with Decision Transformer (Meng et al., 2023), we train the critic using $\lambda$-returns rather than discounted Monte Carlo returns, and store the action probabilities at sampling time instead of recomputing them during training.

## 4.2 MAIN RESULTS

Table 1 reports the normalized returns and standard deviations over three random seeds for each method. Overall, our GRPO-DT achieves the best performance across most tasks. PPO-DT also

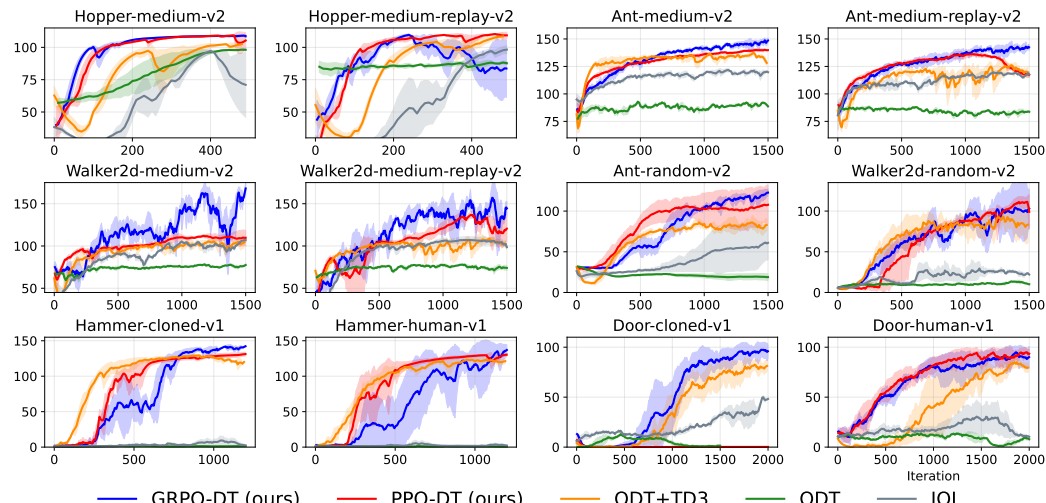

Figure 2: Results on part of the environments and datasets. Our adapted GRPO-DT perform the best on most of the environments and dataset. ODT+TD3 and PPO-DT yield competitive results on most of the environments while ODT and IQL keeps converge on local optimum.

performs competitively in many cases. ODT+TD3 obtains reasonable results, while ODT and IQL consistently underperform, particularly on tasks with low-quality pretraining data such as the *random* datasets and on challenging domains like Adroit. Note that as we perform longer training iterations as mentioned in Section 4.1, the results for ODT+TD3 are better than the reported ones from the original paper (Yan et al., 2024).

**Low offline data quality.** The first part in Table 1 shows results when pretrained with offline data of low quality. We observe that both our adapted GRPO-DT and PPO-DT perform significantly better on *random* datasets. Since these datasets consist of trajectories generated by an untrained random policy, pretraining on them initializes the agent with poor or even harmful biases, often causing the policy to collapse or converge to suboptimal solutions. Our results indicate that GRPO-DT and PPO-DT exhibit stronger robustness to such low-quality pretraining, achieving superior asymptotic performance compared to baselines. In contrast, ODT—relying purely on supervised learning signals—fails to escape local optima, and IQL suffers from similar limitations.

**Medium data quality.** The rest parts of Table 1 present results when pretrained with offline data of decent quality. For the **MuJoCo environments**, our GRPO-DT and PPO-DT achieves best results while ODT+TD3 is competitive and ODT/IQL performs reasonably. In **Adroit**, where state and action spaces are substantially larger and more complex, policies are highly prone to degradation or collapse during finetuning. Under these conditions, ODT and IQL fail to improve pretrained policies, whereas our GRPO-DT consistently achieves high returns, demonstrating strong exploration and stability. ODT+TD3 demonstrates competitive performance on some environments, but falls short of matching the robustness of our approach in some cases. PPO-DT, while strong on some environments, fails to improve on other cases. Training longer or incorporating additional techniques such as reward shaping may alleviate this but we leave it for future work. For **Antmaze** environment where reward is sparse, ODT+TD3 achieves best results while our GRPO-DT performs competitively. Other methods fail to improve the policy.

**Advantages over previous methods.** Our GRPO-DT offers several advantages over prior approaches besides final performance. First, unlike methods that rely on an auxiliary critic, our approach requires no additional networks, making it simpler to implement. Second, by leveraging accurate gradient estimation through sub-trajectory sampling, our method is more computationally efficient, requires much less gradient updates per iteration. For example, our method requires $8 \times 256$ gradient updates per iteration while ODT/ODT+TD3 typically requires $256 \times 300$, much higher than our method. Finally, it can finetune any pretrained DT-style model with minimal modifications

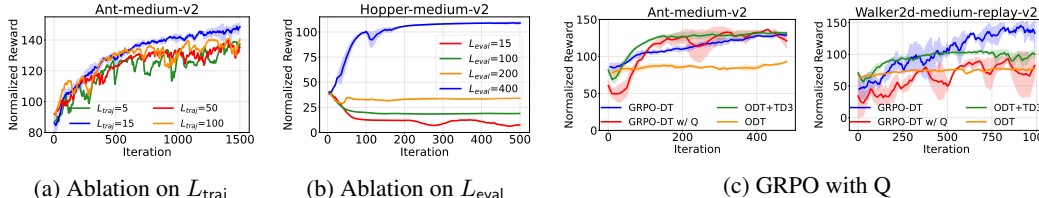

(a) Ablation on $L_{\text{traj}}$     (b) Ablation on $L_{\text{eval}}$     (c) GRPO with Q

Figure 3: Panel (a) shows ablation on sub-trajectory length $L_{\text{traj}}$. Both longer and shorter sub-trajectory length lead to inferior results. Panel (b) shows ablation on evaluation steps $L_{\text{eval}}$. Inadequate evaluation steps lead to model collapse. Panel (c) shows training with our variant described in Algorithm 2. It achieves decent results.

(see Appendix A.5 for experiments), whereas prior methods such as ODT+TD3 require altering the offline pretraining loss to incorporate RL gradients in some cases and training an auxiliary Q-function simultaneously during pretraining, which prevents them from directly finetuning an already pretrained model.

### 4.3 ANALYSES AND ABLATIONS

We provide additional analyses and ablation studies in this section. Empirical evidence supporting the key design choices in Algorithm 1 has been presented in Fig. 1.

**Ablation on sub-trajectory length.** Sub-trajectory in our method represents the unit for assigning advantage. Thus its length is crucial to our algorithm. Empirical results in Fig. 3a confirm that increasing sub-trajectory length destabilizes training and leads to inferior outcomes. However, excessively short sub-trajectories, while stable, also yield sub-optimal results. This is likely because very short trajectories sampled from the same state distribution are overly homogeneous, limiting their ability to provide informative learning signals.

**Ablation on sub-trajectory evaluation steps.** For each sub-trajectory, we extend the rollout with additional evaluation steps ranging from 30 to 400, depending on the environment. As illustrated in Fig. 3b, longer evaluation rollouts enable more reliable assessment of sub-trajectory quality and consequently improve performance.

**Using Q function to replace sub-trajectory generation.** In scenarios where resetting the environment is infeasible, we instead train an auxiliary Q function and apply GRPO with Algorithm 2. As shown in Fig. 3c, this approach still achieves decent performance.

## 5 RELATED WORK

**Transformers for RL.** With transformers becoming the dominant architecture in both CV and NLP, a growing number of transformer-based approaches have been proposed in the RL community (Lin et al., 2023; Chen et al., 2022; Yuan et al., 2024). Owing to their strong capability in modeling sequential dependencies (Parisotto & Salakhutdinov, 2021), transformers are naturally suited for reinforcement learning when formulated as a sequence modeling problem (Chen et al., 2021; Janner et al., 2021; Wang et al., 2022). In this paradigm, models typically condition on past states, actions, and returns to autoregressively predict future actions. However, such approaches rely on offline datasets and often suffer from issues of data scarcity and out-of-distribution problem. This motivates the offline pretraining followed by online finetuning paradigm. Nevertheless, existing works either treat supervised objectives as the primary training signal when tuning transformers online (Zheng et al., 2022; Yan et al., 2024), rely on Q-learning rather than transformer-based architectures (Lee et al., 2022; Zheng et al., 2023; Song et al., 2022; Yu & Zhang, 2023; Nair et al., 2020), or are situated in MARL settings (Meng et al., 2023). In contrast, our work focuses on online finetuning of offline-pretrained decision-making transformers using purely RL-based gradients.

**RL for transformers.** Reinforcement learning has also emerged as a powerful technique for aligning and enhancing large language models (LLMs) (Ouyang et al., 2022; Lee et al., 2023). A wide

spectrum of algorithms has been explored, ranging from policy gradient methods such as PPO, to off-policy methods like Implicit Language Q-Learning (ILQL) (Snell et al., 2022) and VerifierQ (Qi et al., 2024), as well as reward-model-free methods such as DPO (Rafailov et al., 2023) and KTO (Ethayarajh et al., 2024). More recently, novel algorithms such as GRPO and approaches like ReFT (Luong et al., 2024) have been proposed to further improve the reasoning ability of LLMs. RL methods have also been applied to transformer-based multi-modal models (Liu et al., 2025; Shen et al., 2025). However, the strategies designed for training LLMs cannot be directly transferred to finetuning Decision Transformers, as decision-making tasks fundamentally differ from language generation in terms of environment dynamics, reward distributions, and optimization objectives. To this end, our work adapts RL algorithms widely adopted in LLMs, specifically GRPO and PPO, to the context of finetuning Decision Transformers.

## 6    CONCLUSION

We presented a systematic study on applying pure RL gradients for online finetuning of Decision Transformers. We identified hindsight return relabeling as the key obstacle for methods featuring importance ratio, and introduced GRPO‐DT with modifications including sub-trajectory training, environment resetting, and sequence-level importance ratios to enable critic-free and efficient finetuning of pretrained DT-style models. In addition, we implemented PPO for DTs (PPO‐DT), showing that pure RL gradients in online stage substantially improve DTs across diverse benchmarks.

**Limitations and future work.**    While effective, our methods assume environment resetting and may face challenges in sparse-reward or very long-horizon tasks. Moreover, long rollouts slows down the training process especially when evaluation steps are relatively long. Our method also requires extensive hyperparameter tuning when deployed to a new environment. Future work includes developing reset-free strategies, scaling to more complex domains, and combining our approach with stronger architectures and exploration techniques to further enhance robustness and generalization.

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

## A APPENDIX

### A.1 ENVIRONMENTAL AND DATASET DETAILS

#### A.1.1 MUJOCO ENVIRONMENTS

We conduct our experiment on three MuJoCo environments:

- **Hopper.** Hopper is a MuJoCo-based single-legged locomotion task where the agent controls three joints to make the robot hop forward while maintaining stability. The action space is 3-dimensional continuous, corresponding to torques applied at the joints, each bounded within $[-1, 1]$. The observation space has 11 dimensions, consisting of positional and velocity information. At each timestep, the reward is a combination of survival bonus, forward progress, and a control cost penalty proportional to the squared magnitude of the action. Episodes terminate when the agent falls or reaches the maximum horizon (default 1000 steps).

- **Walker2d.** Walker2D is a 2D bipedal walking robot task where the agent controls six joints to make the robot walk forward steadily. The action space is a 6-dimensional continuous vector (torques in $[-1, 1]$) applied to hinge joints. The observation space has 17 dimensions. At each timestep, the agent receives a reward composed of (i) a "healthy" survival bonus, (ii) a forward progress reward proportional to the displacement in the x-direction, and (iii) a control cost penalty proportional to the magnitude of the action. Episodes terminate if the robot becomes unhealthy (e.g. torso height out of range, non-finite states) or reaches the maximum horizon.

- **Ant.** The Ant task is a 3-dimensional locomotion problem where the agent controls an 8-joint quadruped to move forward while maintaining balance. The action space is an 8-dimensional continuous vector (typically bounded in $[-1, 1]$). The observation space comprises the robot's positional and velocity state (and sometimes contact observations). Each timestep the agent receives a reward combining a forward-progress term (displacement in the x-axis), a control cost penalty (proportional to the squared action magnitude), and often an alive bonus. Episodes terminate when the ant falls or the time horizon (default 1000) is reached.

The size and normalized return of each offline dataset is presented in Table 2.

Table 2: The size and normalized rewards of offline datasets used in MuJoCo environments.

| Dataset | Size | Normalized Reward |
|---|---|---|
| Hopper-medium-v2 | 999906 | 44.32±12.27 |
| Hopper-medium-replay-v2 | 402000 | 14.98±16.32 |
| Hopper-random-v2 | 999906 | 1.19±1.16 |
| Walker2d-medium-v2 | 999995 | 62.09±23.83 |
| Walker2d-medium-replay-v2 | 302000 | 14.84±19.48 |
| Walker2d-random-v2 | 999997 | 0.01±0.09 |
| Ant-medium-v2 | 999946 | 80.30±35.82 |
| Ant-medium-replay-v2 | 302000 | 30.95±31.66 |
| Ant-random-v2 | 999930 | 6.36±10.07 |

#### A.1.2 ADROIT ENVIRONMENT

We choose three Adroit environments to experiment:

- **Door.** The Door task requires a 28-DoF hand-arm system to unlatch and open a door. The action space is 28-dimensional continuous, with each joint command scaled to $[-1, 1]$ The observation space has 39 dimensions, including joint states, latch status, and relative positions between the hand and handle. The dense reward combines distance penalties, velocity regularization, and bonuses for increasing door hinge displacement, encouraging successful door opening.

- **Hammer.** The Hammer task involves a 28-DoF robotic hand-arm system (a 24-DoF ShadowHand plus a 4-DoF arm) that must pick up a hammer and drive a nail into a board. The action space is 26-dimensional continuous, representing joint commands (scaled into $[-1, 1]$). The observation space is 46-dimensional, encoding joint states, poses of the hammer and nail, and forces on the nail. The reward combines terms for progress in driving the nail (hinge displacement or insertion depth), penalties on control magnitude, and distance-based cost.

- **Pen.** The Pen task requires a 24-degree-of-freedom robotic hand to manipulate a pen into a target orientation. The action space is 24-dimensional continuous, with joint commands scaled to $[-1, 1]$ for each actuator. The observation space is 45-dimensional, including joint states, pen pose, and the goal orientation. The reward is composed of a negative penalty proportional to the Euclidean distance between the pen and target, an orientation similarity term (dot product between real and target orientation), proximity bonuses when both distance and angular alignment are sufficiently tight, and a dropping penalty if the pen falls.

The corresponding offline dataset quality can be found in Table 3.

Table 3: The size and normalized rewards of offline dataset used in Adroit environment.

| Dataset | Size | Normalized Reward |
|---|---|---|
| Pen-cloned-v1 | 499886 | 108.63± 122.43 |
| Pen-human-v1 | 4800 | 202.69± 154.48 |
| Hammer-cloned-v1 | 999872 | 8.11± 23.35 |
| Hammer-human-v1 | 10948 | 23.80± 33.36 |
| Door-cloned-v1 | 999939 | 12.29± 18.35 |
| Door-human-v1 | 6504 | 28.35± 13.88 |

## A.2 ANTMAZE ENVIRONMENT

The Umaze environment in Antmaze places an Ant quadruped in a U-shaped maze. The action space is 8-dimensional continuous, with torques in $[-1, 1]$. The observation space is a goal-aware dictionary: a 27-dimensional "observation" vector (positions and velocities of the Ant body parts), plus 2D achieved goal and desired goal vectors indicating the Ant's torso position and the target goal in the plane. The reward provide is sparse: 0 if the ant hasn't reached its final target position, and 1 if the ant is in the final target position (the ant is considered to have reached the goal if the Euclidean distance between both is lower than 0.5 m). The quality of the offline datasets used is presented in Table 4.

Table 4: The size and the average and standard deviation of the normalized reward of the Antmaze datasets used in our experiments.

| Dataset | Size | Normalized Reward |
|---|---|---|
| Antmaze-Umaze-v2 | 998573 | 86.14± 34.55 |
| Antmaze-Umaze-Diverse-v2 | 999000 | 3.48± 18.32 |

## A.3 EXPERIMENTAL DETAILS

### A.3.1 HYPERPARAMETERS

Table 5 shows the hyperparameters that are common across all our experiments and Table 6 summarizes the domain-specific hyperparameters for each environment and dataset for GRPO-DT. For antmaze-environment, following ODT+TD3's (Yan et al., 2024) practice, We remove all 1-step trajectories, because the size of the replay buffer for decision transformers is controlled by the number of trajectories, and antmaze dataset contains a large number of 1-step trajectories due to its data generation mechanism (immediately terminate an episode when the agent is close to the goal, but do

not reset the agent location). And we did not add positional embedding as suggested by ODT (Zheng et al., 2022).

For GRPO–DT, we collect 1 complete trajectory for replay buffer per iteration in MuJoCo and Antmaze environments and 5 complete trajectories each iteration in Adroit environments. The buffer size for the complete trajectories is 32. When doing resetting, we sample 16 trajectories from the complete trajectories buffer. We choose four reset points for each trajectory and the group size for each trajectory is 8. This results in 512 sub-trajectories per iteration. The buffer size for this sub-trajectories is 2048.

For PPO–DT, we collect 8 trajectories for MuJoCo and Antmaze environment and 8 or 16 trajectories for Adroit each iteration. The buffer size is 4 times of the number of trajectories collected per iteration. Following ODT+TD3's practice, we add Layernorm (Ba et al., 2016) to the critic of PPO–DT in Adroit and Antmaze environment to stabilize training process. Table 5 summarizes the architecture used in PPO–DT, and additional environment-specific hyperparameters appear in Appendix A.3.1.

For the Q-function-guided GRPO–DT, we conduct experiments on *Ant-medium-v2* and *Walker2d-medium-replay-v2*. We generally follow the hyperparameter settings of ODT+TD3 for training the Q-functions. Specifically, the critic learning rate is set to $1 \times 10^{-3}$, the discount factor $\gamma$ is 0.99, the policy noise has mean $0$ and standard deviation $0.1$, and is clipped within $(-0.5, 0.5)$. The target critic and policy are updated with a step size of $0.005$. For each state, we sample $64$ actions from the predicted policy distribution and assign rewards to them using the learned Q-function. The advantages of GRPO–DT are then computed within each action group. The policy learning rate is $1 \times 10^{-3}$, with a KL coefficient of $0.001$ and an entropy coefficient of $0.01$.

For ODT+TD3 and ODT baselines, we use their original code and parameter setting respectively. For IQL baseline, we generally follow ODT+TD3's implementation, but set pretraining steps to the same as other baselines in our experiments for fair comparison.

Table 5: The common hyperparameters in our experiments.

| Hyperparameters | Value |
|---|---|
| Number of layers | 4 |
| Number of attention heads | 4 |
| Embedding dimension | 512 |
| Actor Optimizer | LAMB (You et al., 2019) |
| Dropout | 0.1 when pretraining, disabled when finetuning |
| Nonlinearity function | SiLU (Elfwing et al., 2018) |
| Weight decay | 0.0001 |
| Gradient norm clip | 0.5 |
| Target entropy | -dim($\mathcal{A}$) |
| PPO Critic layer | 2 |
| PPO Critic hidden size | 256 for Mujoco, 512 for others |
| PPO Critic activation | SiLU |
| PPO Critic Optimizer | AdamW (Loshchilov & Hutter, 2017) |

## A.4 GRPO WITH Q FUNCTION

In this section we introduce GRPO with Q, an action-level variant of our method designed for settings where environment resets are infeasible. Instead of generating multiple sub-trajectories from the same state, our method samples a group of actions under the current policy for each visited state and evaluates them with an auxiliary Q-function. The resulting Q-values are normalized to provide advantages, which are then used to optimize the policy via the GRPO objective. Meanwhile, the Q-function is updated following standard TD3 practice. This design preserves the core idea of group-based policy optimization while eliminating the need for environment reset.

## A.5 TRAINING WITH OTHER ARCHITECTURE

To evaluate the generality of our algorithm, we further apply it to other DT-style architectures. *Reinformer* (Zhuang et al., 2024) is a max-return sequence modeling approach for offline reinforcement

Table 6: The hyperparameters that we use to finetune DT with GRPO-DT in each domain, where $T_{train}$ and $T_{eval}$ stands for context length for training and evaluation, $\gamma$ is the discount factor, $lr_a$ represents learning rate for the actor, $L_{traj}$ and $L_{eval}$ represent sub-trajectory length and evaluation steps for each sub-trajectory respectively, $\varepsilon$ is Clipping threshold, $\varepsilon_{GRPO}$ is the minimum deviation of a sub-trajectory's raw reward from the mean reward of its group, ETPR is the initial entropy temperature for online finetuning.

| Environ | BS | $T_{train}$ | $T_{eval}$ | RTG | $\gamma$ | $lr_a$ | $L_{traj}$ | $L_{eval}$ | $\varepsilon$ | $\varepsilon_{GRPO}$ | ETPR |
|---|---|---|---|---|---|---|---|---|---|---|---|
| Ho-M(R) | 256 | 20 | 1 | 7200 | 0.995 | 5e-5 | 15 | 400 | 0.2 | 2.0 | 0.20 |
| Ho-R | 256 | 20 | 1 | 7200 | 0.995 | 5e-5 | 15 | 400 | 0.2 | 2.0 | 0.20 |
| Wa-M(R) | 256 | 20 | 1 | 10000 | 0.995 | 5e-5 | 15 | 400 | 0.3 | 2.0 | 0.04 |
| Wa-R | 256 | 20 | 1 | 10000 | 0.995 | 5e-5 | 15 | 400 | 0.3 | 2.0 | 0.20 |
| An-M(R) | 256 | 20 | 1 | 12000 | 0.995 | 5e-5 | 15 | 200 | 0.3 | 2.0 | 0.04 |
| An-R | 256 | 20 | 1 | 12000 | 0.995 | 5e-5 | 15 | 200 | 0.3 | 2.0 | 0.20 |
| D-C | 512 | 5 | 1 | 3000 | 0.99 | 3e-5 | 10 | 100 | 0.3 | 0.5 | 0.10 |
| D-H | 512 | 5 | 1 | 3000 | 0.99 | 3e-5 | 10 | 100 | 0.3 | 0.4 | 0.04 |
| P-C | 512 | 5 | 1 | 6000 | 0.99 | 3e-5 | 3 | 30 | 0.3 | 0 | 0.02 |
| P-H | 512 | 5 | 1 | 6000 | 0.99 | 3e-5 | 3 | 30 | 0.3 | 0 | 0.02 |
| H-C | 512 | 5 | 5 | 4000 | 0.99 | 3e-5 | 10 | 100 | 0.3 | 0 | 0.05 |
| H-H | 512 | 5 | 5 | 4000 | 0.99 | 3e-5 | 10 | 100 | 0.3 | 0.8 | 0.05 |
| U | 256 | 5 | 1 | 2 | 1.0 | 5e-5 | 10 | 200 | 0.2 | 0 | 0.05 |
| UD | 256 | 1 | 5 | 2 | 1.0 | 5e-5 | 10 | 200 | 0.2 | 0 | 0.05 |

Table 7: The hyperparameters that we use to finetune DT in each domain with PPO-DT, where $CL_{train}$ and $CL_{eval}$ stands for context length for training and evaluation, $lr_a$ represents learning rate for the actor and $lr_c$ is the learning rate for the critic, $n_{PPO}$ is the number of online trajectories sampled each iteration, ETPR is the initial entropy temperature for online finetuning. The discount factor $\gamma$ is 0.99, clipping range parameter $\varepsilon$ is 0.2 and GAE-$\lambda$ is 0.95.

| Environ | BS | $CL_{train}$ | $CL_{eval}$ | RTG | $lr_c$ | $lr_a$ | $n_{PPO}$ | ETPR |
|---|---|---|---|---|---|---|---|---|
| Ho-M(R) | 256 | 20 | 1 | 7200 | 1e-3 | 5e-5 | 8 | 0.02 |
| Ho-R | 256 | 20 | 1 | 7200 | 1e-3 | 5e-5 | 8 | 0.04 |
| Wa-M(R) | 256 | 20 | 1 | 10000 | 1e-3 | 5e-5 | 8 | 0.02 |
| Wa-R | 256 | 20 | 1 | 10000 | 1e-3 | 5e-5 | 8 | 0.20 |
| An-M(R) | 256 | 20 | 1 | 12000 | 1e-3 | 5e-5 | 8 | 0.02 |
| An-R | 256 | 20 | 1 | 12000 | 1e-3 | 5e-5 | 8 | 0.02 |
| D-C | 512 | 5 | 1 | 3000 | 2e-4 | 3e-5 | 16 | 0.002 |
| D-H | 512 | 5 | 1 | 3000 | 2e-4 | 3e-5 | 16 | 0.002 |
| P-C | 512 | 5 | 1 | 6000 | 2e-4 | 3e-5 | 8 | 0.04 |
| P-H | 512 | 5 | 1 | 6000 | 2e-4 | 3e-5 | 8 | 0.04 |
| H-C | 512 | 5 | 5 | 4000 | 2e-4 | 3e-5 | 16 | 0.005 |
| H-H | 512 | 5 | 5 | 4000 | 2e-4 | 3e-5 | 16 | 0.005 |
| U | 256 | 5 | 1 | 2 | 1e-3 | 5e-5 | 8 | 0.02 |
| UD | 256 | 1 | 5 | 2 | 1e-3 | 5e-5 | 8 | 0.02 |

learning. It integrates the RL objective of return maximization into supervised sequence modeling by using expectile regression to predict the in-distribution maximum return, which then guides optimal action generation. This method enhances trajectory stitching capability and achieves state-of-the-art performance among sequence models on the D4RL benchmark, particularly on tasks requiring learning from suboptimal data. The training process of applying GRPO-DT to this architecture is presented in Fig. 4.

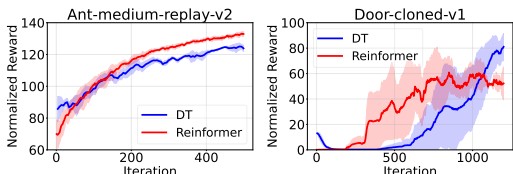

Figure 4: Applying our GRPO-DT to *Reinformer*

---

**Algorithm 2** GRPO with Q (action-level variant)

---

**Input:** Pretrained policy $\pi_\theta$, trajectory buffer $\mathcal{T}_{\text{replay}}$, auxiliary Q-function $Q_\phi$, total rounds $T$, group size $G$, discount factor $\gamma$.

1: **for** round $t = 1, \cdots, T$ **do**
2:    Rollout trajectory $\tau$ using current policy $\pi_\theta(\cdot|s, g)$; update $\mathcal{T}_{\text{replay}}$ with $\tau$. **// Trajectory collection with FIFO buffer update.**
3:    Sample a minibatch $\mathcal{G}$ from $\mathcal{T}_{\text{replay}}$ with probability $p(\tau) \propto |\tau|$.
4:    **for** each $\tau \in \mathcal{G}$ **do**
5:       For each state $s_h$ in $\tau$, sample $G$ actions $\{a_{h,i}\}_{i=1}^G \sim \pi_\theta(\cdot|s_h, g_h)$.
6:       Evaluate each sampled action with $Q_\phi(s_h, a_{h,i})$.
7:       Normalize scores $\{Q_\phi(s_h, a_{h,i})\}$ to obtain advantages $\{\widehat{A}_{h,i}\}$. **// Action-level evaluation with Q-function.**
8:       Update policy $\pi_\theta$ using GRPO objective with advantages $\{\widehat{A}_{h,i}\}$.
9:       Update $Q_\phi$ following TD3-style critic learning.

---

## A.6 THE USE OF LARGE LANGUAGE MODELS (LLMS)

LLMs were used to polish the writing of this paper.

