# OpenReview forum: "Online Finetuning Decision Transformers with Policy Gradients"
_ICLR.cc/2026/Conference — Submitted to ICLR 2026_

### Official Review · Reviewer_pGmC · 2025-10-23

**Soundness:** 3
**Presentation:** 4
**Contribution:** 3
**Rating:** 6
**Confidence:** 4

**Summary:**

This paper proposes a novel online finetuning method for Decision Transformer (DT) that uses only GRPO gradients for update.
To adapt GRPO to DT, the authors propose several modifications to GRPO: 1) update based on intended Returns-To-Go (RTG) to
avoid instability in importance sampling caused by RTG being off-policy; 2) update GRPO based on subtrajectories. To calculate
advantages within a group, several reset points are sampled along the trajectory based on the action variance (if reset is not possible,
a TD3-based critic is used for advantage estimation instead); 3) the importance ratio term is calculated sequence-wise to ensuring consistency between the objective and the advantage signal. On several environments, the proposed method outperforms baselines such
as ODT, IQL and TD3+ODT.

**Strengths:**

1. The paper is well-written and easy to follow. The motivation is clearly stated in the first page of the paper, followed by clearly listed contribution and paper organization section which helps a lot for readers to grasp the general idea. There are also summaries that helps understanding for sections such as Sec. 3.2 and Sec. 4; overall, the paper possesses a good layout. The pseudocode also helps for understanding the paper.

2. The proposed idea is well-motivated and intuitive. I particularly like the idea of sampling reset points with respect to variance, which reflects the insight of "crucial tokens" [1] in today's RL for LLM community. The sequence-level importance ratio also reflects recent improvements on GRPO, such as GSPO [2].

3. The empirical results is solid; the proposed method is tested on many environments, including very challenging ones such as Adroit. The detailed hyperparameter listed in Tab. 5 and 6 enhances the reproducibility of the paper. It is exciting that the proposed method is also applicable to alternative architectures such as Reinformer as suggested in Fig. 4.

4. I appreciate that the authors provide a fallback for environments that are unable to reset, which addresses a large concern in the RL community.

**References**

[1] S. Wang et al. Beyond the 80/20 Rule: High-Entropy Minority Tokens Drive Effective Reinforcement Learning for LLM Reasoning. ArXiv: 2506.01939, 2025.

[2] C. Zheng et al. Group Sequence Policy Optimization. ArXiv: 2507.18071, 2025.

**Weaknesses:**

As shown in Fig. 3 and Tab. 6, In order to make the algorithm to work well on many MuJoCo environments, the $L_{\text{eval}}$ needs to be very long (400 steps), which significantly slows down the training process. The method could also potentially struggle with sparse reward / goal-based scenario where the reward is rarely gained in the middle of the episode.

**Minor Weaknesses**

There are several typos throughout the paper. To list a few:

1. The full name of GRPO and PPO are not introduced in the first use of abbreviation;

2. line 50, graidents -> gradients;

3. Fig. 1 b), And -> and; the caption misses a period.

4. Tab. 5, the caption misses a period.

5. line 84 misses a period.

6. Fig. 3 b), the legend should be $L_{\text{eval}}$ instead of $L_{\text{traj}}$.

**Questions:**

I have several questions:

1. While I agree that "prioritizing supervised learning" is a mindset brought by decision transformer itself that worths breaking, there is also a possible middle ground before "prioritizing supervised learning" and "pure RL": RL-led training (i.e. much smaller supervised learning coefficient for ODT+TD3, or adding a small supervised coefficient for the proposed GRPO finetuning). In fact, the unification of post-training stages (RL+SFT) is also adopted by many recent LLM papers [1, 2, 3]. I think this is also a potential baseline worth trying.

2. In line 370, the authors mention "Atari" environment. Why is Atari results not appearing in the paper? (There is only one appearance of "Atari" in the paper besides author names in the reference list.)

3. Why does GRPO and QGRPO have very different initial performance before training in Fig. 3 c)?

4. Could the authors specify line 317-318, "using $\lambda$-returns rather than discounted Monte Carlo returns"? Does this mean PPO uses TD($\lambda$)? If so, what is the value of $\lambda$ (which is not shown in Tab. 5)? This $\lambda$ seems to confllict with the entropy term coefficient $\lambda$ in Eq. 3.

I am open to increase my score if the concerns raised in the question and weakness section can be well-addressed.

**References**

[1] X. Lv et al. Towards a Unified View of Large Language Model Post-Training. ArXiv: 2509.04419, 2025.

[2] J. Yan et al. LUFFY: Learning to Reason Under Off‑Policy Guidance. In NeurIPS, 2025.

[3] L. Ma et al. Learning What Reinforcement Learning Can't: Interleaved Online Fine-Tuning for Hardest Questions. ArXiv: 2506.07527, 2025.

---

> ### Author Response · Authors · 2025-12-02
> **Author response (1/2)**
>
> Dear Reviewer,
>
> Thank you for taking the time to review our paper. Below, we provide detailed responses and clarifications. We have also updated the manuscript accordingly.
>
> **Response to Weaknesses:**
>
> >**Weakness:** As shown in Fig. 3 and Tab. 6, In order to make the algorithm to work well on many MuJoCo environments, the $L_{eval}$  needs to be very long (400 steps), which significantly slows down the training process. The method could also potentially struggle with sparse reward / goal-based scenario where the reward is rarely gained in the middle of the episode.
>
> **Response:** Thank you for your comment. We would like to clarify that although a larger $L_{eval}$ increases the cost of rollout collection, it **does not cause substantial slowdown in overall training**. The reason is that more accurate return estimates allow GRPO-DT to require **only 4–8 gradient updates per iteration**, which is two orders of magnitude fewer than temporal-difference-based methods such as ODT+TD3. Thus, the moderate additional rollout cost is largely offset by dramatically reduced update cost.
>
> Regarding sparse or goal-based environments, we did not observe performance degradation. In fact, our method performs competitively on AntMaze, a standard sparse-reward benchmark that requires long-horizon goal-reaching.
>
> This robustness comes from our design that **separates sub-trajectory rollouts (of length $L_{traj}$) from return-estimation rollouts (of length $L_{eval}$)**. A small $L_{traj}$ provides fine-grained credit assignment, while a larger $L_{eval}$ provides stable long-horizon return estimates.
>
> >**Minor Weakness:** There are several typos throughout the paper. To list a few…
>
> **Response:** Thank you for pointing out these typos. We have corrected all the mentioned typos, and have additionally performed a thorough pass over the entire manuscript to ensure consistency and clarity.
>
> **Response to Questions:**
>
> >**Question 1:** While I agree that "prioritizing supervised learning" is a mindset brought by decision transformer itself that worths breaking, there is also a possible middle ground before "prioritizing supervised learning" and "pure RL": RL-led training (i.e. much smaller supervised learning coefficient for ODT+TD3, or adding a small supervised coefficient for the proposed GRPO finetuning). In fact, the unification of post-training stages (RL+SFT) is also adopted by many recent LLM papers [1, 2, 3]. I think this is also a potential baseline worth trying.
>
> **Response:** Thank you for your thoughtful question. We agree that a hybrid RL-led + light supervised finetuning strategy is an interesting direction.
>
> However, applying such a hybrid objective to **Decision Transformers (DTs)** is non-trivial due to a fundamental incompatibility rooted in **hindsight return relabeling**. As shown in the ODT work [1], hindsight relabeling is *required* for supervised sequence modeling (see their Fig. 5.4). In contrast, our analysis in Section 3.1 demonstrates that **hindsight relabeling is incompatible with the importance ratios required in policy gradient methods**, including PPO and GRPO.
>
> Due to this inherent incompatibility, hybrid supervision cannot be straightforwardly applied without redesigning the relabeling mechanism itself. Addressing this conflict lies beyond the scope of our current submission, and we view it as promising future work.
>
> [1] Zheng, Qinqing, Amy Zhang, and Aditya Grover. "Online decision transformer." ICML 2022.
>
> >**Question 2:** In line 370, the authors mention "Atari" environment. Why is Atari results not appearing in the paper? (There is only one appearance of "Atari" in the paper besides author names in the reference list.)
>
> **Response:** Thank you for catching this typo. The “Atari” on line 370 was a typo and should be changed to “Adroit”, which is one of the environments used in our experiments. We have corrected this typo, and have additionally performed a thorough pass over the entire manuscript to ensure consistency and clarity.
>
> >**Question 3:** Why does GRPO and QGRPO have very different initial performance before training in Fig. 3 c)?
>
> **Response:** Thank you for the question. The performance difference between GRPO and QGRPO at the beginning of Fig. 3c is due to our evaluation protocol: evaluation at each iteration is conducted **after** the gradient updates of that iteration. Thus, even at iteration 0, both methods have already undergone several updates, during which their behaviors may diverge.
>
> Additionally, some environments are highly sensitive to small policy changes and rollout stochasticity in the early stage, which can further amplify these differences. Together, these factors explain the initial performance gap observed between GRPO and QGRPO.

---

> > ### Author Response · Authors · 2025-12-02
> > **Author response (2/2)**
> >
> > >**Question 4:** Could the authors specify line 317-318, "using lambda -returns rather than discounted Monte Carlo returns"? Does this mean PPO uses TD(lambda)? If so, what is the value of lambda (which is not shown in Tab. 5)? This  seems to conflict with the entropy term coefficient lambda  in Eq. 3.
> >
> >
> > **Response:** Thank you for your question. Yes, our PPO adaption uses $TD(\lambda)$ to estimate advantages, following the standard generalized advantage estimation (GAE) formulation [2]. The value of $\lambda$ is 0.95 in all our experiments. We have clarified this in the revised manuscript and added the exact hyperparameter setting to the Appendix.
> >
> > To avoid confusion between the GAE parameter $\lambda$ and the entropy regularization coefficient, we have renamed the entropy coefficient to $\kappa$ in the updated version.
> >
> > [2] Schulman, John, et al. "High-dimensional continuous control using generalized advantage estimation." arXiv preprint 2015.

---

### Official Review · Reviewer_FUMy · 2025-11-01

**Soundness:** 3
**Presentation:** 3
**Contribution:** 2
**Rating:** 4
**Confidence:** 3

**Summary:**

This paper addresses a central challenge in extending Decision Transformers (DT) to online reinforcement learning settings: how to perform online finetuning with pure RL gradients rather than supervised losses. The authors observe that existing approaches (e.g., ODT, ODT + TD3) rely heavily on supervised objectives and that the use of hindsight return relabeling breaks on-policy importance sampling. They propose a novel adaptation of the GRPO algorithm to Decision Transformers for critic-free finetuning. The method introduces four key components: (i) sub-trajectory optimization to reduce variance, (ii) consistent state resetting for grouped rollouts, (iii) sequence-level importance ratios to align with advantage definitions, and (iv) an active selection mechanism that focuses updates on high-uncertainty states. Extensive experiments on D4RL benchmarks show that this method outperforms ODT, ODT+TD3, and IQL across MuJoCo, Adroit, and AntMaze tasks, especially when offline data is of low quality.

**Strengths:**

1. Novel Contribution: The proposed GRPO bridges policy gradient methods with ranking-based loss, enabling DT finetuning without requiring a Q-function or supervised pretraining loss modification.
2. Critic-Free Optimization: By using actual rollout returns and relative advantage estimation, the method avoids instability and estimation bias common in value-based methods.
3. High Compatibility: GRPO can be applied to pretrained DTs with minimal architecture changes, offering plug-and-play flexibility.
4. Action Variance-Based Reset Sampling: This active sampling strategy focuses optimization on uncertain states, potentially improving learning efficiency.
5. Extensive Evaluation: Experiments span multiple environments and dataset qualities (medium, expert, random), with results showing strong average performance and robustness, especially in low-quality offline settings.

**Weaknesses:**

1. Motivational concerns: The paper presents critic-free finetuning as its main contribution, citing instability and complexity of value-based methods. However, this motivation appears incompletely grounded: 1) No theoretical or empirical evidence is provided to substantiate that critics (e.g., in TD3) cause consistent instability. 2) Recent DT-based methods have successfully employed critics without major issues. 3) The GRPO approach incurs significantly higher sample cost, shifting rather than solving the valuation problem. 4) In environments where resets are not possible, GRPO resorts to using value functions, contradicting the motivation.
Therefore, while the method is interesting, its justification for replacing value-based learning is not sufficiently convincing.

2. Strong Reset Assumption: The method assumes the ability to reset environments to arbitrary past states, which is not feasible in many real-world applications or simulators.
3. Disaster Risk from High-Variance States: Resetting to states with high action variance can lead to rollout trajectories with consistently poor rewards. This may cause catastrophic forgetting, as GRPO updates policy based on the best among poor trajectories.
4. Loss of Temporal Coherence: Using short sub-trajectories breaks the global sequence modelling that DT is known for, potentially harming long-horizon decision-making performance.
5. Relative Advantage Lacks Global Signal: The group-based normalized advantage only provides relative ranking within a local batch, and may reinforce suboptimal behaviours if the entire group performs poorly.
6. High Sample Complexity: GRPO requires a large number of rollouts per iteration (G × K × G), which leads to high environment interaction costs compared to alternatives like TD3.

**Questions:**

1. Can you clarify what specific failure modes or inefficiencies of critics your method addresses that are not already mitigated in prior work?
2. How do you ensure that the selected reset points actually lead to meaningful learning rather than reinforcing poor behaviours or inducing catastrophic forgetting?
3. Is there a risk that all trajectories in a group are poor, and yet one is still “reinforced” due to its relative return?
4. Have you observed any degradation in performance on tasks that require long-horizon credit assignment?

---

> ### Author Response · Authors · 2025-12-02
> **Author response (1/3)**
>
> Dear Reviewer,
>
> Thank you for taking the time to review our paper. Below, we provide detailed responses and clarifications. We have also updated the manuscript accordingly.
>
> **Response to Weaknesses:**
>
> >**Weakness 1:** Motivational concerns: The paper presents critic-free finetuning as its main contribution, citing instability and complexity of value-based methods. However, this motivation appears incompletely grounded: 1) No theoretical or empirical evidence is provided to substantiate that critics (e.g., in TD3) cause consistent instability. 2) Recent DT-based methods have successfully employed critics without major issues. 3) The GRPO approach incurs significantly higher sample cost, shifting rather than solving the valuation problem. 4) In environments where resets are not possible, GRPO resorts to using value functions, contradicting the motivation. Therefore, while the method is interesting, its justification for replacing value-based learning is not sufficiently convincing.
>
> **Response:** Thank you for your comment. We believe part of this concern stems from a misunderstanding of the motivation behind our work. Our goal is not to argue that critic-based RL should be avoided—in fact, we also provide a critic-enabled variant of our method for settings without environment resets.
>
> Our motivation is different: existing online DT methods (e.g., ODT and ODT+TD3) continue to rely heavily on supervised sequence-modeling losses, even during online interaction, which limits their ability to adapt and improve policies.
>
> This motivates our central question: Can DTs be finetuned online using **pure RL gradients**—analogous to recent successes of RL-based finetuning in LLMs?
>
> Our results show that the answer is **yes**. A key insight is that **hindsight return relabeling**, widely used in online DT methods, is incompatible with the importance ratios required in policy gradient algorithms; **removing it is essential for stable RL finetuning** (Section 3.1). Building on this, we introduce several crucial design components—**sub-trajectory rollouts for improved credit assignment, sequence-level importance ratios for stability, and active sampling for exploration** (Section 3.2).
>
> With these modifications, our GRPO-DT algorithm **successfully enables pure-RL online finetuning of DTs and achieves new state-of-the-art performance across diverse environments and tasks**.
>
> >**Weakness 2:** Strong Reset Assumption: The method assumes the ability to reset environments to arbitrary past states, which is not feasible in many real-world applications or simulators.
>
> **Response:** Thank you for your comment. While we agree that arbitrary resetting is not always feasible, environment resets are supported in many important domains—including perfect-information games (e.g., Go, Chess), LLM reasoning tasks [1], and widely used simulator-based RL benchmarks [2]. Recent theoretical and empirical work also shows that incorporating reset operations can substantially improve sample efficiency and policy performance in online RL [2,3]. Our method follows this established line of work.
>
> Crucially, resetting is what allows **our GRPO-DT algorithm to achieve new state-of-the-art performance across multiple environments and tasks**, by enabling sub-trajectory rollouts that provide significantly improved credit assignment.
>
> That said, **resetting is not a strict requirement of our approach**. For settings where resets are infeasible, we also develop a Q-function-guided GRPO-DT variant that replaces resetting with value-based action evaluation. As shown in Fig. 3c, this variant still achieves strong performance, demonstrating that GRPO-DT remains effective in reset-free environments.
>
> [1] Kazemnejad, Amirhossein, et al. "VinePPO: Refining Credit Assignment in RL Training of LLMs." ICML 2024.
>
> [2] Mhammedi, Zak, Dylan J. Foster, and Alexander Rakhlin. "The power of resets in online reinforcement learning." NeurIPS 2024.
>
> [3] Yin, Dong, et al. "Sample efficient deep reinforcement learning via local planning." arXiv preprint arXiv:2301.12579.

---

> ### Author Response · Authors · 2025-12-02
> **Author response (2/3)**
>
> >**Weakness 3:** Disaster Risk from High-Variance States: Resetting to states with high action variance can lead to rollout trajectories with consistently poor rewards. This may cause catastrophic forgetting, as GRPO updates policy based on the best among poor trajectories.
>
> **Response:** Thank you for your comment. High-variance states typically correspond to regions where the policy is most uncertain or under-trained. Improving policy on these states is beneficial and aligns with prior findings showing that prioritizing uncertain regions can accelerate policy improvement [3]. This is also reflected in your Strength #4.
>
> Empirically, we observe exactly this effect: actively selecting such states leads to clear gains. As shown in Fig. 1e, our active selection mechanism substantially outperforms variants without it. This indicates that our algorithm effectively leverages these challenging states rather than being harmed by them.
>
> [3] Yin, Dong, et al. "Sample efficient deep reinforcement learning via local planning." arXiv preprint arXiv:2301.12579.
>
> >**Weakness 4:** Loss of Temporal Coherence: Using short sub-trajectories breaks the global sequence modelling that DT is known for, potentially harming long-horizon decision-making performance.
>
> **Response:** Thank you for your comment. We would like to clarify that using short sub-trajectories **does not break the temporal coherence of pretrained Decision Transformers**. In fact, **the original DT formulation never trains on full trajectories**: it is trained on short, randomly sampled sub-sequences from long trajectories. This is the standard sequence-modeling paradigm used in DT and all subsequent DT-based methods.
>
> Our online finetuning procedure follows the same design. Using short sub-trajectories therefore **preserves---rather than disrupts---the temporal inductive biases of DT**. It aligns the finetuning regime with the pretraining regime and additionally enables finer-grained credit assignment than full-trajectory GRPO. Our empirical results across multiple environments further validate that this design choice does not harm long-horizon performance.
>
> >**Weakness 5:** Relative Advantage Lacks Global Signal: The group-based normalized advantage only provides relative ranking within a local batch, and may reinforce suboptimal behaviours if the entire group performs poorly.
>
> **Response:**  Thank you for the comment. We agree that group-normalized advantages provide only relative signals within each batch—**this is a general property of GRPO itself rather than a limitation specific to our adaptation**. In practice, however, relative advantages are sufficient as long as batches contain meaningful variation: even modest positive signals drive iterative improvement. This behavior is well-validated in prior applications of GRPO to LLM reasoning, and empirically, **our GRPO-DT achieves state-of-the-art performance across multiple environments and tasks**, demonstrating the effectiveness of leveraging group-based learning signals.
>
> >**Weakness 6:** High Sample Complexity: GRPO requires a large number of rollouts per iteration (G × K × G), which leads to high environment interaction costs compared to alternatives like TD3.
>
> **Response:** Thank you for your comment. We agree that GRPO-style policy gradient methods naturally require more environment interactions per iteration than value-based methods such as TD3. This reflects a well-known trade-off in RL: **policy gradient methods typically use more samples, while temporal-difference methods require substantially more gradient updates and higher computational overhead**.
>
> In our setting, this trade-off is clearly favorable. Despite higher sample usage, GRPO-DT requires **two orders of magnitude fewer gradient updates** and no auxiliary critics, making each iteration computationally lighter. More importantly, **GRPO-DT consistently improves upon existing baselines and achieves state-of-the-art performance across multiple environments and tasks**, demonstrating that the added samples translate into meaningful and robust performance gains.
>
> Finally, we emphasize that all baselines in our experiments are trained for extended online iterations to ensure they reach their best possible performance. As shown in Fig. 2, temporal-difference methods tend to plateau early, indicating they are unlikely to benefit further from additional online interactions.

---

> > ### Author Response · Authors · 2025-12-02
> > **Author response (3/3)**
> >
> > **Response to Questions:**
> >
> > >**Question 1:** Can you clarify what specific failure modes or inefficiencies of critics your method addresses that are not already mitigated in prior work?
> >
> > **Response:** Please see our response to Weakness 1, where we provide detailed clarification on the motivation behind our method. To reiterate, our goal is not to argue against the use of critic-based RL, but rather to develop the **first online finetuning algorithm for Decision Transformers that operates with pure RL gradients**. This requires addressing limitations specific to prior DT-based methods (e.g., reliance on hindsight relabeling), rather than issues with critics.
> >
> > >**Question 2:** How do you ensure that the selected reset points actually lead to meaningful learning rather than reinforcing poor behaviours or inducing catastrophic forgetting?
> >
> > **Response:** Please see our response to Weakness 3.
> >
> > >**Question 3:** Is there a risk that all trajectories in a group are poor, and yet one is still “reinforced” due to its relative return?
> >
> > **Response:** Please see our response to Weakness 5.
> >
> > >**Question 4:** Have you observed any degradation in performance on tasks that require long-horizon credit assignment?
> >
> > **Response:** We did not observe any degradation in tasks requiring long-horizon credit assignment. In particular, our method performs competitively on AntMaze, a benchmark that requires long-range dependency and multi-step reasoning.
> >
> > This robustness comes from our design that **separates sub-trajectory rollouts (of length $L_{traj}$) from return-estimation rollouts (of length $L_{eval}$)**. A small $L_{traj}$ provides fine-grained credit assignment, while a larger $L_{eval}$ provides stable long-horizon return estimates—effectively preserving long-horizon signals even though optimization occurs on short sub-trajectories.

---

### Official Review · Reviewer_3pRc · 2025-11-01

**Soundness:** 2
**Presentation:** 2
**Contribution:** 2
**Rating:** 4
**Confidence:** 3

**Summary:**

This paper focuses on the online fine-tuning of Decision Transformer (DT). It first points out that the "hindsight return relabeling" mechanism in traditional ODT leads to inconsistency between the RTG in online interaction and training phases, undermining the optimization foundation of importance sampling.
Then, it proposes the core solution: retaining the original target RTG of online trajectories to eliminate this relabeling mechanism.
On this basis, the paper adapts the GRPO to the online fine-tuning scenario of DT, and puts forward four key modifications: sub-trajectory sampling, environment resetting, sequence-level importance ratio, and active selection.

**Strengths:**

The analysis of hindsight return relabeling is remarkably clear and thorough, with notably effective results from the associated ablation studies.The paper features a well-structured framework and high readability.

**Weaknesses:**

1. The motivation of this paper requires further elaboration. The adoption of pure RL gradients in LLMs stems from the extremely high memory overhead and training costs associated with value functions. However, this issue does not exist in offline reinforcement learning (offline RL).
2. The exploration of online fine-tuning capabilities lacks investigation into concatenation ability, particularly in environments such as AntMaze.
3. The performance of certain baselines is significantly lower—for instance, the results of IQL on AntMaze UD-v2 and P-C-v1.

**Questions:**

1. Is the explanation for Figure 1 missing? What do subfigures (c), (d), and (e) represent respectively?
2. What is the unit of the x-axis in Figure 2? And why does IQL start from scratch in the Hopper-Medium-v2 and Hopper-Medium-Replay-v2 environments?

---

> ### Author Response · Authors · 2025-12-02
> **Author response (1/2)**
>
> Dear Reviewer,
>
> Thank you for taking the time to review our paper. Below, we provide detailed responses and clarifications. We have also updated the manuscript accordingly.
>
> **Response to Weaknesses:**
>
> >**Weakness 1:** The motivation of this paper requires further elaboration. The adoption of pure RL gradients in LLMs stems from the extremely high memory overhead and training costs associated with value functions. However, this issue does not exist in offline reinforcement learning (offline RL).
>
> **Response:** Thank you for the comment. We’d like to clarify that our work specifically studies **online finetuning of Decision Transformers (DTs)**—a setting fundamentally different from offline RL. Existing online DT methods (e.g., ODT and ODT+TD3) continue to rely heavily on supervised sequence-modeling losses, even during online interaction, which limits their ability to adapt and improve policies.
>
> This motivates our central question: Can DTs be finetuned online using **pure RL gradients**—analogous to recent successes of RL-based finetuning in LLMs?
>
> Our results show that the answer is **yes**. A key insight is that **hindsight return relabeling**, widely used in online DT methods, is incompatible with the importance ratios required in policy gradient algorithms; removing it is essential for stable RL finetuning (Section 3.1). Building on this, we introduce several crucial design components—**sub-trajectory rollouts for improved credit assignment, sequence-level importance ratios for stability, and active sampling for exploration** (Section 3.2).
>
> With these modifications, our GRPO-DT algorithm **successfully enables pure-RL online finetuning of DTs and achieves new state-of-the-art performance across diverse environments and tasks**.`
>
> >**Weakness 2:** The exploration of online fine-tuning capabilities lacks investigation into concatenation ability, particularly in environments such as AntMaze.
>
> **Response:** Thank you for the comment. We agree that evaluating concatenation ability is an interesting direction. However, this is orthogonal to the main focus of our work. Our contribution is to provide the **first online DT method that finetunes purely with RL gradients**, enabled by several key design choices (e.g., removing hindsight return relabeling, sub-trajectory rollouts, sequence-level importance ratios). As shown in our experiments, **our algorithm achieves new state-of-the-art performance across multiple environments and tasks**. We view analyzing concatenation ability as a promising extension for future work.
>
> >**Weakness 3:** The performance of certain baselines is significantly lower—for instance, the results of IQL on AntMaze UD-v2 and P-C-v1.
>
> **Response:** Thank you for your comment. We first clarify that our IQL result on P-C-v1 is comparable with the results reported in the ODT+TD3 paper [1].
>
> For AntMaze UD-v2, although our IQL score is lower than what is shown in [1], the IQL baseline in [1] was pretrained for **1M steps**, which is substantially larger than the number of pretraining steps used for other baselines in their paper (typically 40k or 5k steps, depending on the environment). In our experiments, we pretrain IQL using the *same* number of steps as all other methods to ensure a fair comparison. We also reran the        IQL experiments multiple times and consistently obtained similar results, so we believe our reported numbers are correct and reliable.
>
> Importantly, the **relative ordering of baselines remains consistent with [1]**—IQL < ODT+TD3—and we further show that ODT+TD3 < our method. We also note that our ODT+TD3 results are generally slightly higher than those reported in [1], as we train all methods with more online interaction steps.
>
> [1] Yan, Kai, Alex Schwing, and Yu-Xiong Wang. "Reinforcement learning gradients as vitamin for online finetuning decision transformers." NeurIPS 2024.

---

> > ### Author Response · Authors · 2025-12-02
> > **Author response (2/2)**
> >
> > **Response to Questions:**
> >
> > >**Question 1:** Is the explanation for Figure 1 missing? What do subfigures (c), (d), and (e) represent respectively?
> >
> > **Response:** Thank you for pointing this out. We apologize for the confusion. We have updated the caption of Figure 1 in the revised manuscript.
> >
> > **Question 2:** What is the unit of the x-axis in Figure 2? And why does IQL start from scratch in the Hopper-Medium-v2 and Hopper-Medium-Replay-v2 environments?
> >
> > **Response:** The x-axis in Figure 2 denotes the number of training iterations, as explained in Section 4.1.
> >
> > Regarding IQL: it does **not** learn from scratch. We report performance **after each iteration** of online finetuning, and on Hopper-Medium-v2 and Hopper-Medium-Replay-v2, the performance of IQL after the first finetuning iteration happens to be lower than that of the other methods. This may give the impression that it starts from scratch.
> >
> > Additionally, the y-axis for these two environments **doesn’t start from 0**, with the goal of better visualizing differences among methods that achieve higher returns. IQL’s initial performance happens to be low, so it is not visible at the start of the curve. We have adjusted the figures to better visualize the initial performance of IQL.

---

### Official Review · Reviewer_2jVM · 2025-11-04

**Soundness:** 2
**Presentation:** 2
**Contribution:** 2
**Rating:** 4
**Confidence:** 4

**Summary:**

This paper proposes an online decision transformer method based purely on policy gradients. The key techniques include optimization on sub-trajectories, providing consistent states, sequence-level importance ratio, and active selection. Experiments on locomotion and manipulation tasks validate the effectiveness of the proposed method.

**Strengths:**

- Improving the online decision transformer is, in my view, an interesting and important direction, and this paper focuses on that topic.

**Weaknesses:**

- Requiring multiple environment resets at each policy update step is clearly a major drawback. In most real-world or complex simulation environments, it’s almost impossible to reset the system to an arbitrary state, which makes the proposed method impractical in such settings.
- I believe the technique of optimization on sub-trajectories is potentially problematic. GRPO removes the critic mainly because the rewards are extremely sparse in outcome-reward setting, which makes value estimation unreliable. As a result, it assigns the same advantage to all tokens. However, in your tasks the rewards are dense, so giving all sub-trajectories the same advantage wastes valuable reward information. I’m not sure what the benefit of using GRPO is in a dense-reward setting like this.
- In addition, the technical details of optimization on sub-trajectories are unclear. What exactly does "taking mean actions for another $L_{eval}$ steps" mean? Why do you distinguish between $L_{traj}$ and $L_{eval}$? Why not all sample on-policy?
- The baselines should include *online off-policy* algorithms such as SAC, instead of only IQL, which mainly belongs to the *offline* RL domain.

I think this paper still needs substantial clarification and is clearly not ready for acceptance at its current stage.

**Questions:**

See above

---

> ### Author Response · Authors · 2025-12-02
> **Author response (1/2)**
>
> Dear Reviewer,
>
> Thank you for taking the time to review our paper. Below, we provide detailed responses and clarifications. We have also updated the manuscript accordingly.
>
> **Response to Weaknesses:**
>
> >**Weakness 1:** Requiring multiple environment resets at each policy update step is clearly a major drawback. In most real-world or complex simulation environments, it’s almost impossible to reset the system to an arbitrary state, which makes the proposed method impractical in such settings.
>
> **Response:** Thank you for your comment. While we agree that arbitrary resetting is not always feasible, environment resets are supported in many important domains—including perfect-information games (e.g., Go, Chess), LLM reasoning tasks [1], and widely used simulator-based RL benchmarks [2]. Recent theoretical and empirical work also shows that incorporating reset operations can substantially improve sample efficiency and policy performance in online RL [2,3]. Our method follows this established line of work.
>
> Crucially, resetting is what allows **our GRPO-DT algorithm to achieve new state-of-the-art performance across multiple environments and tasks**, by enabling sub-trajectory rollouts that provide significantly improved credit assignment.
>
> That said, **resetting is not a strict requirement of our approach**. For settings where resets are infeasible, we also develop a Q-function-guided GRPO-DT variant that replaces resetting with value-based action evaluation. As shown in Fig. 3c, this variant still achieves strong performance, demonstrating that GRPO-DT remains effective in reset-free environments.
>
> [1] Kazemnejad, Amirhossein, et al. "VinePPO: Refining Credit Assignment in RL Training of LLMs." ICML 2025.
>
> [2] Mhammedi, Zak, Dylan J. Foster, and Alexander Rakhlin. "The power of resets in online reinforcement learning." NeurIPS 2024.
>
> [3] Yin, Dong, et al. "Sample efficient deep reinforcement learning via local planning." arXiv preprint arXiv:2301.12579.
>
> >**Weakness 2:** I believe the technique of optimization on sub-trajectories is potentially problematic. GRPO removes the critic mainly because the rewards are extremely sparse in outcome-reward setting, which makes value estimation unreliable. As a result, it assigns the same advantage to all tokens. However, in your tasks the rewards are dense, so giving all sub-trajectories the same advantage wastes valuable reward information. I’m not sure what the benefit of using GRPO is in a dense-reward setting like this.
>
> **Response:** Thank you for the thoughtful comment. We first clarify the complementary strengths and weaknesses of GRPO and PPO [1]:
>
> - **GRPO** provides accurate reward estimation and a simple, critic-free implementation, but suffers from poor credit assignment, since full-trajectory GRPO assigns the same advantage to all tokens.
> - **PPO** provides fine-grained token-level credit assignment, but suffers from noisy reward estimation and instability due to value-function training.
>
> Our **sub-trajectory rollout mechanism is precisely designed to combine the strengths of both approaches**. Instead of assigning advantages to the entire trajectory (as in vanilla GRPO), we:
>
> - perform **credit assignment at the sub-trajectory level**,
> - control the **granularity of credit assignment** via the sub-trajectory length, and
> - avoid training an additional value network.
>
> As shown in Fig. 1b, this design substantially improves performance over standard GRPO, demonstrating that sub-trajectory rollout effectively improves credit assignments. **This improvement is key to why our GRPO-DT achieves new state-of-the-art performance across multiple environments and tasks.**
>
> [1] Kazemnejad, Amirhossein, et al. "VinePPO: Refining Credit Assignment in RL Training of LLMs." ICML 2025.

---

> ### Author Response · Authors · 2025-12-02
> **Author response (2/2)**
>
> >**Weakness 3:** In addition, the technical details of optimization on sub-trajectories are unclear. What exactly does "taking mean actions for another $L_{eval}$ steps" mean? Why do you distinguish between $L_{traj}$ and $L_{eval}$? Why not all sample on-policy?
>
> **Response:** Thank you for your question. We first clarify the two action generation modes used in our GRPO-DT design:
>
> - **Sub-trajectory rollout ($L_{traj}$ steps).** Sample actions based on the current policy for $L_{traj}$ steps. This sampled segment is what GRPO optimizes against and thus **serves as the unit of credit assignment**.
> - **Return estimation rollout ($L_{eval}$ steps).** After the sub-trajectory is generated, we extend the rollout for an additional $L_{eval}$ steps using the **expected action** under the policy distribution (or the most probable action in the discrete case). These additional $L_{eval}$ steps are only used to estimate the return of the sub-trajectory, but otherwise don't directly participate in GRPO optimization.
>
> We separate these two horizons because they serve fundamentally different purposes:
>
> - $L_{traj}$ determines the *granularity* of credit assignment and must be **on-policy** (as in standard GRPO).
> -  $L_{eval}$ controls the *quality* of return estimation, and using the expected action substantially **reduces variance** compared to fully sampled on-policy rollouts.
>
> This separation allows GRPO-DT to retain on-policy correctness for optimization while improving stability through low-variance return estimation.
>
> >**Weakness 4:** The baselines should include online off-policy algorithms such as SAC, instead of only IQL, which mainly belongs to the offline RL domain.
>
> **Response:** Thank you for your comment. We’d like to clarify that, as stated in Section 4.1, we use **the online variant of IQL**, which performs online policy improvement. This baseline choice follows the practice of the recent ODT+TD3 paper [4], ensuring consistency with prior work on online finetuning of Decision Transformers.
>
> [4] Yan, Kai, Alex Schwing, and Yu-Xiong Wang. "Reinforcement learning gradients as vitamin for online finetuning decision transformers." NeurIPS 2024.

---

### Meta-Review · Area_Chair_rZBQ · 2026-01-09

**Summary:**

The reviewers' primary reservations centered on the practicality and motivation of the proposed method, specifically regarding the reliance on environment resets. A recurring and significant concern across multiple reviews was that the requirement to reset environments to arbitrary states for sub-trajectory rollouts is impractical for many real-world applications and complex simulators, limiting the method's utility compared to standard online RL. Additionally, reviewers questioned the fundamental motivation for prioritizing a critic-free, pure policy gradient approach, arguing that the instability of critics was not sufficiently proven to justify the high sample complexity and potential loss of global temporal coherence introduced by the proposed GRPO adaptation. Finally, there were specific concerns regarding the strength and fairness of baselines, particularly the performance of IQL and the exclusion of other off-policy algorithms, alongside questions about whether the method truly outperforms hybrid RL and supervised learning approaches.

**Reviewer Concerns:**

The authors provided a robust rebuttal that effectively addressed technical misunderstandings and baseline discrepancies, particularly regarding the configuration of the IQL baseline and the incompatibility of hindsight return relabeling with importance sampling, which satisfactorily explained why a simple hybrid supervised loss was not used. The clarification on the "Atari" typo and the justification for the evaluation horizon (LevalLeval​) versus update costs also resolved specific queries from Reviewer pGmC. However, the concern regarding the environment reset assumption remains a significant outstanding issue; while the authors successfully argued that this is an established practice in specific research domains (like games and LLMs) and allows for better credit assignment, the fundamental critique that this design choice restricts the algorithm's broader applicability compared to standard actor-critic methods was defended but not fully resolved for reviewers prioritizing generalizable real-world RL. Similarly, the trade-off involving high sample complexity due to the GRPO formulation remains a valid point of contention that the rebuttal acknowledged but framed as a necessary cost for stability.

**Reviewer Scores:**

Reviewer pGmC, who was already positive, would likely raise their score to a 8 or remain 6, as their specific questions regarding evaluation horizons, typos, and the "Atari" confusion were clearly resolved. Reviewer 3pRc would likely increase their score to a 6, given that their concerns about baseline fairness and the distinct motivation for online finetuning versus offline RL were directly addressed with evidence. Reviewers 2jVM and FUMy might be more hesitant; while the authors clarified the mechanics of sub-trajectory optimization and pointed out logical inconsistencies in FUMy’s review regarding active sampling, the fundamental disagreement regarding the practicality of the reset assumption and the high sample complexity might prevent them from fully endorsing the paper, though they would likely acknowledge the empirical state-of-the-art results within the defined constraints.

---

### Decision · Program_Chairs · 2026-01-26

Reject